# Green Synthesis of Silver Nanoparticles and Its Combination with *Pyropia columbina* (Rhodophyta) Extracts for a Cosmeceutical Application

**DOI:** 10.3390/nano13061010

**Published:** 2023-03-10

**Authors:** Mercedes González-Conde, Julia Vega, Félix López-Figueroa, Miguel García-Castro, Ana Moscoso, Francisco Sarabia, J. Manuel López-Romero

**Affiliations:** 1Department of Organic Chemistry, Faculty of Sciences, University of Malaga, Campus de Teatinos s/n, 29071 Malaga, Spain; 2Andalusian Institute of Blue Biotechnology and Development (IBYDA) Experimental Center Grice Hutchinson, University of Málaga, Lomas de San Julián, 2, 29004 Malaga, Spain

**Keywords:** silver nanoparticles, green synthesis, daisy, leek, garlic, *Pyropia columbina*, photoprotectors, encapsulation, NIPAM

## Abstract

We report the green synthesis of silver nanoparticles (AgNPs) by using daisy petals (*Bellis perennis*), leek (*Allium porrum*) and garlic skin (*Allium sativum*) as reducing agents and water as solvent. AgNPs are obtained with high monodispersity, spherical shapes and size ranging from 5 to 35 nm and characterized by UV-Vis and TEM techniques. The obtained yields in AgNPs are in concordance with the total phenolic content of each plant. We also study the incorporation of AgNPs in combination with the red algae *Pyropia columbina* extracts (PCE) into cosmetic formulations and analyze their combined effect as photoprotective agents. Moreover, we carry out the inclusion of the PCE containing mycosporine-like amino acids (MAAs), which are strong UV-absorbing and antioxidant compounds, into β-cyclodextrin (βCD) and *p*NIPAM nanoparticles and analyze stability and release. The thermoresponsive polymer is grown by free radical polymerization using *N*-isopropylacrylamide (NIPAM) as the monomer, *N,N*′-methylenebisacrylamide (BIS) as the cross-linker, and 2,2′-azobis(2-methylpropionamidene) (V50) as the initiator, while βCD complex is prepared by heating in water. We evaluate the nanoparticle and βCD complex formation by UV-Vis and FT-IR, and NMR spectroscopies, respectively, and the nanoparticles’ morphology, including particle size, by TEM. The cosmetic formulations are subsequently subjected to accelerated stability tests and photoprotective analyses: a synergistic effect in the combination of AgNPs and PCE in photoprotection was found. It is not related to a UV screen effect but to the antioxidant activity, having potential against photoaging.

## 1. Introduction

Silver nanoparticles (AgNPs) are relevant material due to their biological activities. Consequently, studies have been carried out to define new synthetic routes to improve yields, control sizes and minimize environmental adverse effects, thus giving rise to green synthesis [1]. AgNPs are prepared by two methodologies, the top-down methods, also named “physical methods”, where the size of the silver particles is reduced to the nanoscale by physical methods, such as laser ablation, and bottom-up methods or “chemical methods”, where AgNPs are produced by a solution of a salt in a solvent, followed by the addition of a reducing agent, in a general process that produces Ag(0) from Ag^+^. In chemical methods, a stabilizing agent is added to the solution to prevent the nanoparticles from agglomerating with each other. The particles obtained with this last method have fewer defects and a more homogeneous chemical composition [1].

These methods are costly and environmentally unfriendly. In addition, chemical methods, by using toxic and harmful reagents, produce an increase in the toxicity and reactivity of the AgNPs, which can cause adverse effects on human health [2]. Therefore, use is made of the concept of green or biological chemistry, which can be defined as “the use of chemical techniques and methodologies that reduce or eliminate the use or generation of raw materials, products and byproducts that are hazardous to human health or the environment” [3]. Green synthesis of AgNPs versus standard chemical or physical synthesis has several advantages, for example, non-toxic and recycled reagents are used, environmental contamination is avoided, easy and safe processes are followed, and environmental pollution is also avoided, and the stability of the nanoparticles is usually higher [4].

The method of synthesis through plants is carried out thanks to the presence of certain metabolites in them, which can be classified into primary and secondary metabolites. The primary metabolites are vitamins, reducing sugars, amino acids, proteins, nucleic acids and polysaccharides. The group of secondary metabolites includes alkaloids, terpenes, phenolic compounds, and glycosides, which have a wide variety of properties such as antibacterial, antioxidant and anticarcinogenic [5]. The use of plant cell cultures as a biotechnological tool, on the other hand, is a sustainable alternative that allows the production of sufficient amounts of plant biomass for the ecological synthesis of nanoparticles because it represents a safe, ecological, and clean method. For instance, the biomass of some species such as *Jatropha curcas* [6], *Medicago sativa* [7], *Sesuvium portulacastrum* L. [8], *Cucurbita maxima* [9], *Lycopersicon esculentum* [10], *Catharanthus roseus* [11], *Vitex negundo* [12], and *Randia acuelata* L. [13] has been used to synthesize AgNPs via aqueous extracts of callus culture. Additionally, fungus has also been used for the preparation of AgNPs [14].

Nowadays the area of cosmetics and dermopharmaceuticals is in continuous progress. That is why new advances in dermocosmetics are diverse and cover various objectives such as preventing aging, eliminating hyperpigmentation, or inhibiting sebum production in the skin. One of the developments in the cosmetic and dermopharmaceutical field are the applications of silver nanoparticles, as they are biocompatible and exhibit antimicrobial, antioxidant, anticancer, antiaging and wound healing activities [5]. In addition, these nanoparticles have a considerable surface area to bind a wide variety of components. Consequently, silver and silver-based formulations have been used to treat infections in chronic ulcers and open wounds, as a healing agent or as a protective agent in oxidation and staining.

However, even when AgNPs have been extensively used with antimicrobial purposes, few efforts have been made to analyze its use as an antioxidant, antiaging and photoprotective agent [15,16]. AgNPs were prepared by using *Symphytum officinale* leaves as a reducing extract, showing a significant inhibition of the production of matrix metalloproteinase-1 and IL-6, but at the same time, an increasing effect over the expression of procollagen type 1. These data suggested that AgNPs have photoprotective properties and may have potential to be used as an agent against photoaging [15]. Additionally, the inhibitory effect of AgNPs on photochemical reactions of photosynthesis was investigated using the green alga model *Chlamydomonas reinhardtii* [16].

On the other hand, the study of new bioactive compounds with antioxidant, anti-aging or photoprotective properties in the dermocosmetics sector is also booming due to the disorders generated after sun exposure [17]. The most common disorders following sun exposure are erythema and photoaging. When UVA rays strike the dermis, reactive oxygen species (ROS) are generated due to absorption of the light through skin chromophores, such as DNA, which produce alterations in the expression of certain genes [18], leading to a translation of a series of enzymes that degrade collagen (a process by which there is a loss of flexibility in the skin) and elastin (collagen’s binding molecule) [19].

To avoid these damages, topical photoprotectors can be used. Natural photoprotectors are beginning to be notable in the field of dermocosmetics due to the disadvantages of synthetic ones, which often generate side effects. Due to this fact, the photoprotective properties of natural substances [18], such as those produced by photosynthetic marine organisms, have begun to be studied [19]. These organisms have developed mechanisms of tolerance to UV radiation, since they are highly exposed to the sun in the intertidal system. One of the most relevant mechanisms is the accumulation of mycosporine-like amino acids (MAAs, Figure 1) that can absorb UV radiation acting as photoprotectors and dissipate its energy without the prior formation of ROS showing antioxidant properties. Moreover, they present high photo and thermostability, which are attractive characteristics for the development of cosmeceutical products. These compounds are characterized by having a cyclohexenone or cyclohexenimine chromophore conjugated to the nitrogen substituent of an amino acid [19]. Macroalgae belong to a diverse group of organisms that could be exploited for biomolecule application since MAAs are found in this group in high concentrations, especially in red macroalgae, and mainly in the order Bangiales, as *Porphyra* spp. The content of MAAs can vary depending on the environmental factors, UV radiation being one of the most influential factors [20,21,22]. Despite their potential as active biomolecules, few authors have analyzed the in vitro photoprotective capacity of algae extracts or creams that contained algae extracts [23,24,25], and in vivo UVB-photoprotective activity of macroalgae extracts has been also reported [17]. 

As the first objective of this article, we focus on the synthesis of AgNPs by using plant extracts, in which several factors need to be controlled, such as the initial silver ion concentration, the types and concentration of organic compounds in the crude plant extract, the reaction temperature, the silver source, the pH of the media and the incubation temperature [1]. We have chosen daisy petals (*Bellis perennis*), leek (*Allium porrum*) and garlic skin (*Allium sativum*) as plant materials. To the best of our knowledge, they have not been studied as reducing agents for metal nanoparticles’ preparation. As the second objective, we study the incorporation of combinations of the prepared AgNPs with *Pyropia columbina* extracts (PCE) into cosmetic formulations and analyze their synergic effect as photoprotective agents. Additionally, to improve the stability and solubility in hydrophilic solvents of the actives, we also carry out the inclusion of PCE into β-cyclodextrin (βCD) and *p*-(*N*-isopropylacrylamide) nanoparticles (@*p*NIPAM) and analyze the release properties. The cosmetic formulations are subsequently subjected to accelerated stability tests and photoprotective analyses.

## 2. Materials and Methods

### 2.1. Chemicals and Instrumentation

Reagents were purchased from Merck. For antioxidant measurements ABTS (2,2′-azino-bis(3-ethylbenzothiazoline-6-sulfonic acid) was purchased in Sigma. The water used for all the processes is distilled water, except for obtaining the *p*NIPAM nanoparticles and the encapsulation of the PCE in them, which is Mili-Q water. PCE was supplied by the Andalusian Institute of Blue Biotechnology and Development (IBYDA) from the University of Málaga. *Pyropia columbina* (*PC*) were collected in Concepción (Chile). Algae were transported to the laboratory in a portable fridge at 4 °C, and washed to eliminate debris. Samples were frozen at −80 °C and lyophilized in a LyoQuest, Telstar, or a Scanvac Coolsafe freeze dryer. HPLC analyses were carried out in a 1260 Agilent InfinityLab Series, Santa Clara, CA, USA, equipped with a Luna C8 column from Phenomenex, Aschaffenburg, Germany.

Three plant species were used for the green synthesis of AgNPs. Samples were also collected in the province of Málaga: (i) *Bellis perennis* (commonly known as *Margarita*), only the flower of daisy petals were used; (ii) *Allium sativum* (commonly known as *Garlic*), only garlic peel was used; and *Allium Porrum* (commonly known as *Leek*). In the latter case, each part of the leek has different concentrations of antioxidant compounds, so in this article we used the white and the green parts separately, to finally analyze the differences between them. Body cream base was obtained from Lanette (Gran Velada S.L.), which has isopropyl palmitate, cetearyl alcohol, glyceryl stearate, hydrogenated polyisobutylene, PEG100, glycerin, phenoxyethanol, synthetic beeswax and citric acid as the main components.

For chemical composition analyses of PCE, *p*NIPAM and βCD samples, NMR were recorded in a Bruker Advanced III 500 spectrometer. The ^1^H spectra were recorded at 500 MHz. The compounds were dissolved in deuterated dimethyl sulfoxide (DMSO-d_6_), with residual solvent peaks at δ = 2.50 (DMSO) ppm for ^1^H. For FT-IR analyses of *p*NIPAM samples, a JASCO 6800FV spectrometer was used. The Raman spectra were acquired using a near-infrared (NIR) diode laser at 785 nm (Renishaw inVia Raman spectroscope). For this analysis a small portion of dry βCD samples was placed on a glass slide.

For the verification of the size and morphology of the synthesized *p*NIPAM nanoparticles, samples were suspended in methanol and deposited on a copper grid or wafer. Once the solvent evaporated, after 24 h, the sample was analyzed, and images were collected in a JEM1400 TEM microscope operated at 100 kV.

Formation of the AgNPs was confirmed by recording the UV-Vis spectra. The surface plasmon band for AgNPs appears around 410–490 nm [26]. To carry out the absorbance measurement, a UV-Vis spectrophotometer Libra S22, with 190–1100 nm of bandwidth, <3 nm resolution, operating with a Xe lamp, and with deionized water as the blank, was used.

Photoprotection factors were measured by using a spectrophotometer UV-2600, Shimadzu, Duisburg, Germany, equipped with an integrating sphere (Shimadzu ISR 2600 Plus, Madrid, Spain). Polymethyl methacrylate plates (PMMA), with a surface of 25 × 25 mm and a roughness of 6 μm (Schönberg, Hamburg, Germany), were used for the assay.

### 2.2. Preparation of Aqueous Extract (WEOP) of the Different Plant Species

For the preparation of the aqueous extract of the plant species (WEOP) and the synthesis and characterization of the AgNPs, the method described by Yap et al. [27] was followed. In a typical procedure, clean garlic peels (0.5041 g), daisy leaves (0.4253 g) or leek green (1.0178 g) or white (1.0280 g) zones, were placed in flasks and distilled water was added (100 mL). The mixtures were heated at 90 °C for 30 min. After cooling, the solutions were filtered and the filtrates were kept in amber reagent bottles, which were then stored in a refrigerator at 4 °C for future usage.

### 2.3. Green Synthesis of AgNPs Using the WEOP

AgNO_3_ solution (3 mL, 1 M) was added to the prepared WEOP (7 mL), and the mixture was stirred (350 rpm) while heating at 90 °C for 30 min. The change of color was attributed to the reduction of Ag^+^ to Ag^0^ and formation of AgNPs. The solution containing AgNPs was centrifuged at 7000 rpm for 1 h. The transparent solution was separated, while the obtained precipitates were re-dispersed into deionized water and centrifuged again. The process was repeated once. The mixture containing AgNPs was covered with aluminum foil and stored at room temperature (rt) for 24 h prior to the UV-Vis analysis. Finally, the nanoparticle solution was then freeze dried (24 h) to obtain purified AgNPs powder. The dry powder was kept in a sealed vial until further use. Results are shown in Table 1.

### 2.4. Estimation of Total Phenolic Content (TPC) in Plant Species

The total phenolic content of the plant species used in this article was determined by the Folin–Ciocalteu method [28,29]. Gallic acid solutions of 40, 30, 20, 10, 5 and 2.5 mg/L were prepared. An aliquot of each solution (1 mL) was placed in a test tube and diluted with distilled water (10 mL). Then, Folin–Ciocalteu’s reagent (1.5 mL) was added and allowed to incubate at rt for 5 min. After this period, Na_2_CO_3_ (4 mL, 20% *w*/*w*) was added in each test tube, adjusted with distilled water up to the mark of 25 mL, agitated and left to stand for 30 min at rt. Absorbance of the standards was measured at 765 nm using UV-Vis spectrophotometer against blank, distilled water. The same process was followed to analyze the samples. Absorbance of the samples was measured at 765 nm against blank, distilled water [30].

To calculate the total polyphenol content of each sample, the absorption at 765 nm was measured and the following Equation (1) was used [30]:(1)$$TPC\left(\%w/w\right)=\left[\frac{C\cdot V\cdot 10^{-6}}{W}\cdot 100\right]\cdot D$$
where TPC is the total phenolic content expressed in mg gallic acid/g dry extract of plant, C is the concentration of gallic acid, V is the volume of aqueous extract obtained, W is the weight of the plant species we weighed to make the extract, and D is the dilution factor (which in our case is 10). The obtained data are shown in Table 2.

### 2.5. Obtention and Analysis of PCE

For the extraction, dry samples were grounded and extracted in distilled water (10 g of dry biomass in 200 mL of H_2_O) using a blender. After incubation of 2 h in a water bath at 45 °C, extracts were filtered and centrifuged. Finally, extracts were evaporated in a vacuum, frozen at −80 °C and lyophilized.

For the analyses of MAAs, phenolic content and antioxidant capacity, 5 mg of the dry extracts were dissolved in 5 mL of H_2_O. MAAs were quantified using HPLC [21,31]. The dissolved extract was filtered through a 0.2 mm filter and injected in the HPLC system. The detection was performed using a C8 column, applying an isocratic flow of 0.5 mL/min and a mobile phase of 1.5% methanol and 0.15% acetic acid in ultrapure H_2_O. The detection was made using a photodiode array detector at 330 nm. Secondary standards were used for the identification of MAAs, and the quantification was performed using the molar extinction coefficients (ε) of the different MAAs [32]. Results were expressed as mg/g of dry weight (DW). 

Phenols content was determined as described in Section 2.4.

Antioxidant capacity was measured through the ABTS method based on the free radical scavenging capacity. The ABTS assay was performed according to Re et al. with some modifications [33]. The generation of the ABTS radical cation (ABTS^+·^) was performed through the mixing of 7 mM ABTS and 2.45 mM K_2_S_2_O_8_ in phosphate buffer (0.1 M, pH: 7). To ensure the complete formation of the radical, the mixture was stored for 12–16 h at rt. ABTS^+^ solution was diluted with phosphate buffer until the absorbance at 727 nm was around 0.75 ± 0.05, then 50 μL of the extracts were mixed with 950 μL of the diluted ABTS^+^. The mixture was incubated for 8 min at rt and darkness, and absorbances were measured at 727 nm. Trolox was used as standard. 

### 2.6. Encapsulation of PCE in βCD

βCD (420.1 mg, 0.37 mmol) was placed in an Erlenmeyer flask and dissolved in distilled H_2_O (35 mL) by heating at 45 °C under stirring. Once dissolved, without cooling the solution, PCE (494.9 mg) was added. The mixture was stirred for 48 h at 45 °C. After this period, the solution was allowed to cool at rt (26 °C), and the solid was filtered under vacuum and washed with distilled water. Then, the solid was dried in an oven (80 °C) for 48 h, weighed and characterized by Raman spectroscopy and ^1^H-NMR. Preliminary evidence of encapsulation was the precipitation of the complex since it has lower solubility than βCD.

### 2.7. Encapsulation in pNIPAM

The synthesis of polymeric nanoparticles of *N*-isopropylacrylamide (*p*NIPAM) was carried out by adding the monomer NIPAM (340.2 mg, 3.006 mmol) into a flask, and purged in a vacuum line with Ar. Then, *N,N*′-methylenebisacrylamide (46.36 mg, 0.3 mmol), the crosslinking agent that binds the monomers, and water (20 mL) were added. The mixture was stirred during 15 min at 70 °C under Ar atmosphere. After this time, without removing the heat and the agitation, a previously prepared solution of V50, radical initiator responsible for the polymerization process to take place, was added (300 μL, 100 mM). The polymerization reaction took place at 70 °C for 2 h under Ar atmosphere. After this period, the reaction was allowed to cool to rt. Then, four centrifugations were performed for 1 h at 7000 rpm, to precipitate the *p*NIPAM nanoparticles. Finally, the product was freeze-dried for 24 h.

Once the *p*NIPAM nanoparticles were synthesized, the encapsulation of the PCE in them was carried out. For this purpose, @*p*NIPAM (105.3 mg) was weighed in a round bottom flask, Mili-Q water (70 mL) was added, and the solution was heated in a magnetic stirrer with a hotplate at 37 °C. Once dissolved, PCE (1 g) was added and allowed to warm at 37 °C with gentle stirring for 48 h. After this period, three centrifugations of 1 h at 7000 rpm were performed to precipitate the *p*NIPAM nanoparticles with the PCE inside (PCE@ *p*NIPAM). Both the *p*NIPAM and the PCE encapsulated nanoparticles were characterized by ^1^H-NMR and FT-IR. To confirm the entrapment efficacy, the aqueous layer was decanted and then exhaustively extracted with dichloromethane (3 × 5 mL). The dichloromethane extracts were dried over MgSO_4_ and concentrated to dryness. The residue was dissolved in CDCl_3_ and analyzed by ^1^H-NMR.

### 2.8. Cosmetic Formulation

For the preparation of the cosmetic formulations, a body cream base formulation was weighed (1.00 g) in a glass vial (5 mL). Ten samples of each formulation were prepared by adding (i) AgNPs (from 0.1 to 1.0 mg), (ii) PCE (from 0.1 to 1.0 mg), (iii) AgNPs (from 0.1 to 1.0 mg) and PCE (from 1.0 to 10.0 mg), (iv) PCE encapsulated in βCD (from 1.0 to 10.0 mg), (v) PCE encapsulated in βCD (from 1.0 to 10.0 mg) and AgNPs (from 0.1 to 1.0 mg), and (vi) PCE encapsulated *p*NIPAM (from 1.0 to 10.0 mg), to the vial. The mixtures were homogenized by mechanical stirring for 5 min. 

### 2.9. Photoprotection Factor

The different photoprotection factors (sun protection factor(SPF), UV-A protection factor (UVAPF) and other biological effective protection factors (BEPFs)) were determined as described [34,35,36]. Polymethyl methacrylate plates (PMMA), with a surface of 25 × 25 mm and a roughness of 6 μm that simulate the human skin, were used for the assay. 

Two concentrations of the different creams were spread on the PMMA plates at a concentration of 1.2 mg/cm^2^ (32.5 mg on the entire plate), using a fingertip glove (previously saturated with the cream) for no more than 1 min. After 15 min of incubation in darkness and rt, transmittance through the plates were measured. Each cream was spread on three plates, and two measurements per plate were undertaken.

The sun protection factor was calculated using the erythematic action spectrum [37]. For the UVAPF, the persistent pigment darkening action spectrum was used [35]. The BEPFs were also calculated against other action spectra related to UV-A radiation: elastosis [38] and photoaging [39]. The spectra of the different biological effects used were published in Vega et al. [18]. The following Equation (2) was used in all cases:(2)$$\mathrm{BEPFs}=\frac{\int_{\lambda =290}^{\lambda =400}Act.Sp\left(\lambda \right)\times E\left(\lambda \right)\times d(\lambda )}{\int_{\lambda =290}^{\lambda =400}T\left(\lambda \right)\times Act.Sp\left(\lambda \right)\times E\left(\lambda \right)\times d(\lambda )}$$
where, Act.Sp (λ) = action spectra (0–1); E (λ) = spectral irradiance of a sunny midday in summer in Málaga (W/m^2^); D (λ) = wavelength step (1 nm); and T (λ) = transmittance values (0–1).

The photoprotection factors of the formulations that contained PCE extracts were measured again after two months and after ultraviolet (UV) radiation exposure during 1 h (45 W m^−2^; 6% of UV-B and 94% of UV-A), in order to see stability and possible release of the encapsulated compounds.

## 3. Results and Discussion

### 3.1. Preparation of WEOP and AgNPs

For the preparation of the WEOP and the synthesis and characterization of the AgNPs, the method described by Yap et al. [27] was followed. 

The AgNPs’ preparation methodology consists of three stages: (i) reduction reaction, which is possible thanks to the functional groups present in the compounds of the WEOP, and follows by chelation of the Ag^+^, subsequent reduction to Ag^0^ and consequently, change of these functional groups to their oxidized form; (ii) growth of the size of AgNPs; and stabilization step, where the phytocompounds create an envelope around the AgNPs [40]. 

Clean garlic peels, daisy petals, leek green or white zones were extracted with hot water during 30 min at 90 °C. The filtrates (WEOP) were used for the AgNO_3_ reduction by mixing and stirring the mixture during 30 min at 90 °C. The change to yellow-brown is characteristic and indicates the formation of AgNPs; this is due to surface plasmon resonance, which is a size-dependent property of NPs. Then AgNPs were isolated by centrifugation and freeze dried. Yields on AgNPs were good (Table 1) ranging from 36% for leek white zone, to 84% in the case of daisy. 

The formation of AgNPs was confirmed by TEM microscopy (Figure 2) and UV-Vis spectroscopy (Figure 3). The images obtained by the TEM technique of the AgNPs synthesized by using the different plant species confirm that they were successfully prepared and suggest a spherical shape, and a size ranging from 5 to 35 nm (Table 1 and Appendix A).

The UV-Vis spectroscopy was used to analyze the prepared AgNPs and WEOPs (Figure 3). It can be observed that the absorbance peak shifts depend on the starting plant species, and the aqueous extracts present absorbance bands at about 260 and 320 nm (Figure 3a–d, Table 1, λ_max_), associated to the complex mixtures of aromatic, phenolic and aliphatic hydrocarbons present in the plant species [5,41]. Consequently, the plant with the highest content of these compounds is the daisy, since the bands are the most intense (Figure 3a, Table 1, entry 1), on the other hand, due to the weakness of the bands, the plant with the lowest content of aromatic and aliphatic compounds is the white zone of the leek (Figure 3d, Table 1, entry 4).

On the other hand, the AgNPs’ solutions show maxima absorbance bands at about 300 and 440 nm (Table 1, λ_max_), which is characteristic of the surface plasmon of the nanoparticles, and confirms the presence of the nanoparticles in the solution (Figure 3e–h). As can be seen, there is a direct dependence on the yield of production of AgNPs with the aromatic and aliphatic hydrocarbons content in WEOP, reaching higher yields when the presence of hydrocarbon is higher. This fact can be confirmed with the analyses of phenolic components in the WEOP (Table 2).

### 3.2. Estimation of Total Phenolic Content (TPC) in Plant Species 

The total phenolic content of the plant species used in this article was determined by the Folin–Ciocalteu method [28,29]. The basis of this method lies in the ability of the reagent to oxidize to phenolate ions, with the consequent reduction of the heteropolyacid from a +6 oxidation state to a mixture of +6 and +5 oxidation states. This results in a color change in the solution from yellow to blue due to the formation of a molybdenum-tungsten complex which shows an absorption maximum at 765 nm [29]. For phenols to be oxidized, they must be in the form of phenolate ion, so it is necessary to alkalinize the medium for the desired reaction to occur [29]. The obtained data are shown in Table 2.

The mean values of the absorbances are plotted versus the concentration of gallic acid, used as standard, obtaining the calibration curve (r^2^ = 0.9905) included in Appendix A.

The sample with the highest and lowest TPC is the daisy and the white zone of the leek, respectively (Table 2, entries 1 and 4, respectively). These data correlate with the recorded UV-Vis spectra (Figure 3). The difference between the concentrations of the different leek zones can be explained by the environmental conditions in which the leek is found. The green leaves grow above the ground, so they are more exposed to the sun’s rays compared to the white zone. Because of this, the total phenolic content should be higher in the green zone of the leek [42]. The results obtained in this article agree with this fact.

From the data shown in Table 1, the reaction yield is higher in the daisy sample and lower in the leek white zone. This fact agrees with the total phenolic content of the samples: the lower the total phenolic content, the lower the reaction yield should be, since there are fewer antioxidant compounds that make the reduction of Ag^+^ to Ag^0^ possible.

### 3.3. Analysis of PCE

PCE presented 30 mg MAAs per gram of dry extract (DE) which is equal to 6.5 mg/g of dry weight (DW), considering that the yield of the extraction is 20% (Table 3). This value is in accordance with those obtained by other authors in *Porphyra* sp., that varied from 5 to 10.5 mg/g DW [23,24,43]. In general, the phenols content present in algae are lower that the observed in plants. In this work, PCE presented 53.5 mg/g DE (10.7 mg/g DW). In *Porphyra* sp. phenols values varied from 3 to 20 mg/g DW [23,24,35]. PCE also presents a relevant antioxidant capacity (32.5 mg/g DW = 5.8 mmol Trolox Equivalent (TE)/g DW). Among macroalgae, brown algae normally present the highest antioxidant capacity, that used to be positive correlated with the phenols content.

### 3.4. Encapsulation of PCE and Release Analyses

Inclusion of PCE was carried out on βCD and *p*NIPAM. The polymerization reaction of NIPAM has been carried out under free-radical conditions, using an amidinopropane (V50) as an initiator and *N,N*′-methylenbisacrylamide (BIS) as a cross-linker. To optimize several parameters such as particle size, monodispersity and stability, the polymerization reaction was performed at a monomer concentration of 150 mM, while the cross-linker BIS molar percentage was 5% with respect to the monomer. The radical initiator was used in constant concentration (100 mM) in all cases. Under these conditions low particle aggregation and high monodispersity were found [44].

In order to confirm the incorporation of PCE into the polymeric nanoparticles of *p*NIPAM and βCD, NMR analyses were carried out (Figure 4 and Figure 5). Figure 4 shows a comparison of the ^1^H-NMR spectra of PCE encapsulated in *p*NIPAM (Figure 4a), neat *p*NIPAM (Figure 4b) and PCE (Figure 4c). In the spectrum of PCE encapsulated in *p*NIPAM (Figure 4a), the broad peaks associated with *p*NIPAM (Figure 4b) together with the characteristic pattern that it is found in the spectrum of PCE (Figure 4c, marked with blue arrows), including the peak at about 3.56 ppm, can be seen, so we can confirm the presence of PCE in *p*NIPAM nanoparticles.

Moreover, the NMR technique shows values near to total incorporation of PCE into the NPs, since no MAAs signals are observed in the ^1^H-NMR spectra of the extracts of the aqueous layer obtained after centrifugation (see Experimental), and consequently the complete entrapment of *PC* actives by the polymeric matrix. The quantitative incorporation of PCE into the NPs can be explained by the low PCE/polymer relationship during preparation, which is low enough to guarantee the complete incorporation of the PCE into the polymer matrix.

Additionally, FT-IR of PCE@*p*NIPAM samples was also carried out (Appendix A). The presence of *p*NIPAM characteristics bands (1520 and 1625 cm^−1^) combined with the most intense attributed to PCE (1080 cm^−1^) confirms the entrapment of PCE actives by the polymer.

Similarly, the 1H-NMR spectrum of PCE encapsulated in βCD (Figure 5a), neat βCD (Figure 5b), and PCE (Figure 5c) were recorded. As can be observed, the 1H-NMR spectrum of PCE encapsulated in βCD can be considered as the combination of that of βCD and PCE, showing the characteristic peaks of PCE, especially those between 0.5 and 1.5 ppm, confirming the presence of PCE in βCD. Analyses by Raman spectroscopy were also carried out, showing the characteristic shift of the structural vibration of βCD from 477 cm−1 to 487 cm−1, also confirming the formation of the inclusion complex [45].

In vitro release of MAAs from PCE@*p*NIPAM systems was studied by using a cellulose dialysis bag placed in an Erlenmeyer containing a stationary phase PBS buffer solution at pH 7.4, and a constant stirring of 100 rpm. At intervals of 10 min during the first 60 min and every 12 h for a 48 h period, aliquots of 3 mL were withdrawn from the solution. The release medium was replaced with the same volume of PBS. Analyses were carried out at 37 °C to observe the release at body temperature. Drug release was measured by UV spectrophotometry recording the absorption at 322 nm (Appendix A). About 10% of cumulative drug release was reached along the first 60 min, reaching 78% after 48 h. Release of MAAs was found not to be complete after 7 days of storage. This fact can be explained by a strong interaction of the carboxylic and hydroxy groups present in MAAs with the polymeric structure [46,47].

### 3.5. Incorporation of AgNPs and PCE into Cosmetic Formulations

Cosmetic formulations were prepared by the addition of quantities from 0.1 to 1.0 mg of AgNPs, from 0.1 to 10.0 mg of PCE, and from 1.0 to 10.0 mg of encapsulated PCE, to a body cream base (1 g, Figure 6). Complete homogenization was achieved by mechanical stirring (Figure 6), except in the case of PCE encapsulated in *p*NIPAM (Figure 6g), in which lumps remain in the mixture. 

Stability tests were carried out by the storage of samples in sealed vials at 5, 20, 35, and 60 °C for 14 days. The prepared formulations do not show changes in appearance over time, except in the case of the mixture in which neat PCE is incorporated (without the presence of AgNPs). In this sample a color change was observed, from pink (Figure 6b) to yellowish white (Figure 6c) along the first 2 days of storage in all the mentioned temperatures. This fact is worth mentioning since it means that AgNPs play an important role in the cosmetic formulation, promoting the stabilization of MAAs. The stabilization role can be attributed to the antioxidative effect of AgNPs.

### 3.6. Photoprotective Activity

Prepared samples were subjected to transmittance tests to evaluate their photoprotective activity and to obtain their protection factors. Transmittance tests were carried out at the IBYDA laboratory. The UV-Vis spectra and the measured transmittance values of the samples (290-400 nm) are shown in Figure 7. In Figure 7a it can be observed that the presence of AgNPs in the body cream slightly reduces the transmittance in the UV-B band, in the range from 290 to 320 nm. When PCE was included in the cream (Figure 7b), the transmittance was reduced in the UV-A band (from 310 to 360 nm). It is important to note that the combination of both, AgNPs and PCE (Figure 7c), in the formulation of the body cream produces a synergistic effect, showing a reduction of the transmittance values in the UV-A region, increasing the absorption of light up to 35 % when quantities of AgNPs and PCE are 1 and 10 mg, respectively, per gram of body cream, reaching this maximum at 335 nm (Figure 7c). The encapsulated PCE were also incorporated in the extracts and almost no photoprotection effect was observed. These formulations were measured again after two months and were irradiated with UV radiation in order to see possible release of the PCE, but no difference was obtained in any of the cases.

Transmittance values of those creams that showed photoprotection in Figure 7a–c, were translated in different photoprotection factors, which are shown in Table 4. The different calculated photoprotection factors of the base cream were very close to one. Adding 1.0 mg of AgNPs (0.1%) the photoprotection increased (e.g., 1.2 of SPF and 1.16 of UVAPF). Adding 10.0 mg of PCE (1.0%), the photoprotection increased to 1.10 (SPF values) and 1.08 (UVAPF values). When both compounds were added (0.1% of AgNPs and 1.0% of PCE) the SPF was 1.21 and the UVAPF 1.19. In general, the measured photoprotection factors are low, which is in concordance with the low incorporation of the active compounds into the formulated creams, with a maximum of 1%. Moreira et al. added 10 % of *Porphyra* sp. extracts and obtained SPF values of 1.6, due to the narrow peak of the MAAs, that do not cover the UV-B region [25]. Probably, the combination of AgNPs and PCE in a higher combination would increase the photoprotection with a synergistic effect. 

As can be seen in Figure 7, cosmetics formulations including AgNPs and PCE present photoprotection in the areas of the spectrum where the MAAs and AgNPs absorb UV radiation (Figure 7a–c). However, in the samples where the PCE is encapsulated, the low photoprotection effect of MAAs disappears (Figure 7d–f). This fact can be explained by a complete incorporation of the PCE into the massive *p*NIPAM and βCD nanoparticles, avoiding the presence of free residual MAAs in the formulations, and acting as a shield that blocks the light and the absorption of UV radiation by the MAAs. Moreover, this fact is confirmed in the case of PCE@*p*NIPAM, where a slow release of the photoactive compound was found. 

The positive effect of free AgNPs on the PCE photoprotection characteristics can be related to a pickering-like effect. Almeida et al. (2016) [46] reported that green coffee oil and modified starch, and other surfactant-free emulsions, enhanced the protection effect of physical UV filters against UV radiation. Although starch particles presented no intrinsic photoprotection properties, they proved to be a solar protection factor SPF promoter by a synergistic effect [48]. The pickering emulsions concept has become a key strategy for prevention against UV-induced skin damage. Thus, solid particles such as AgNPs could produce a synergetic effect with PCE on photoprotection. This fact can also be attributed to a protective effect against oxidation of the AgNPs over the MAAs present in PCE.

## 4. Conclusions

The synthesis of AgNPs by a green methodology using daisy petals (*Bellis perennis*), leek (*Allium porrum*) and garlic skin (*Allium sativum*) as reducing agents and water as a solvent was conducted. These nanoparticles were characterized by UV-Vis absorption and TEM techniques showing high monodispersity, spherical shapes and size ranging from 5 to 35 nm. The measured total phenolic content of the three plants is in concordance with the reducing properties of the aqueous extracts and the yields AgNPs achieved.

βCD and *p*NIPAM nanoparticles of MAAs actives present in PCE were also prepared and characterized by Raman, FT-IR and NMR techniques. It was found a complete incorporation of the actives into the nanoparticles, and a slow release of MAAs from the PCE@*p*NIPAM system, reaching a maximum of 78% of cumulative release after 48 h. 

Cosmetic formulations were prepared by the addition of combinations of AgNPs and the algae *Pyropia columbina* extracts (PCE) into a base body cream. Formulations show good stability, except in the case of the use of neat PCE, where color changes over the time were found. However, the presence of AgNPs in body cream samples with PCE stabilizes the formulation. This effect can be attributed to an oxidative stabilization of MAAs by the AgNPs. 

The presence of combinations of PCE and AgNPs in the body cream significantly improves the photoprotective properties, reaching values of 1.21 and 1.19 for SPF and UVAPF, respectively. In addition, they may have potential as agents against photoaging. However, the inclusion of the PCE active compounds into βCD and *p*NIPAM nanoparticles inhibit the photoprotective activity of the PCE. More research based on the pickering emulsion design is necessary to improve the photoprotector level of the extract’s formulations and cosmetic creams prepared. 

## Figures and Tables

**Figure 1 nanomaterials-13-01010-f001:**
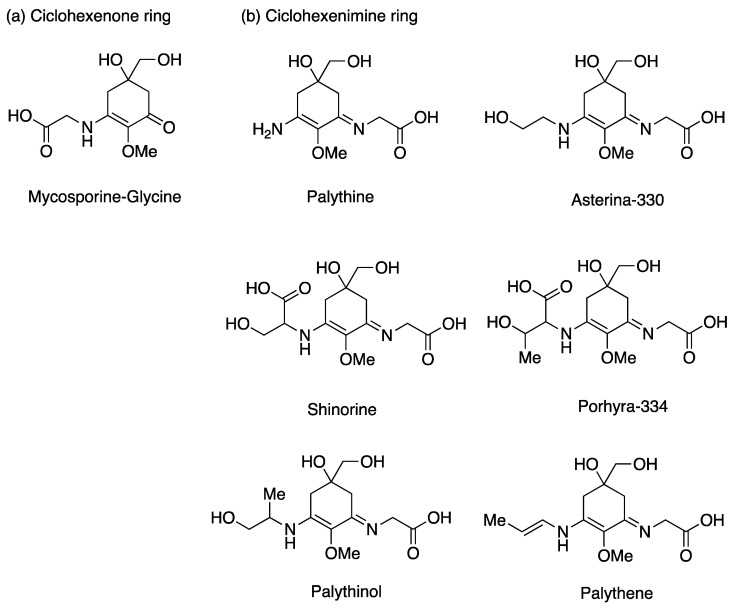
MAAs’ chemical structures.

**Figure 2 nanomaterials-13-01010-f002:**
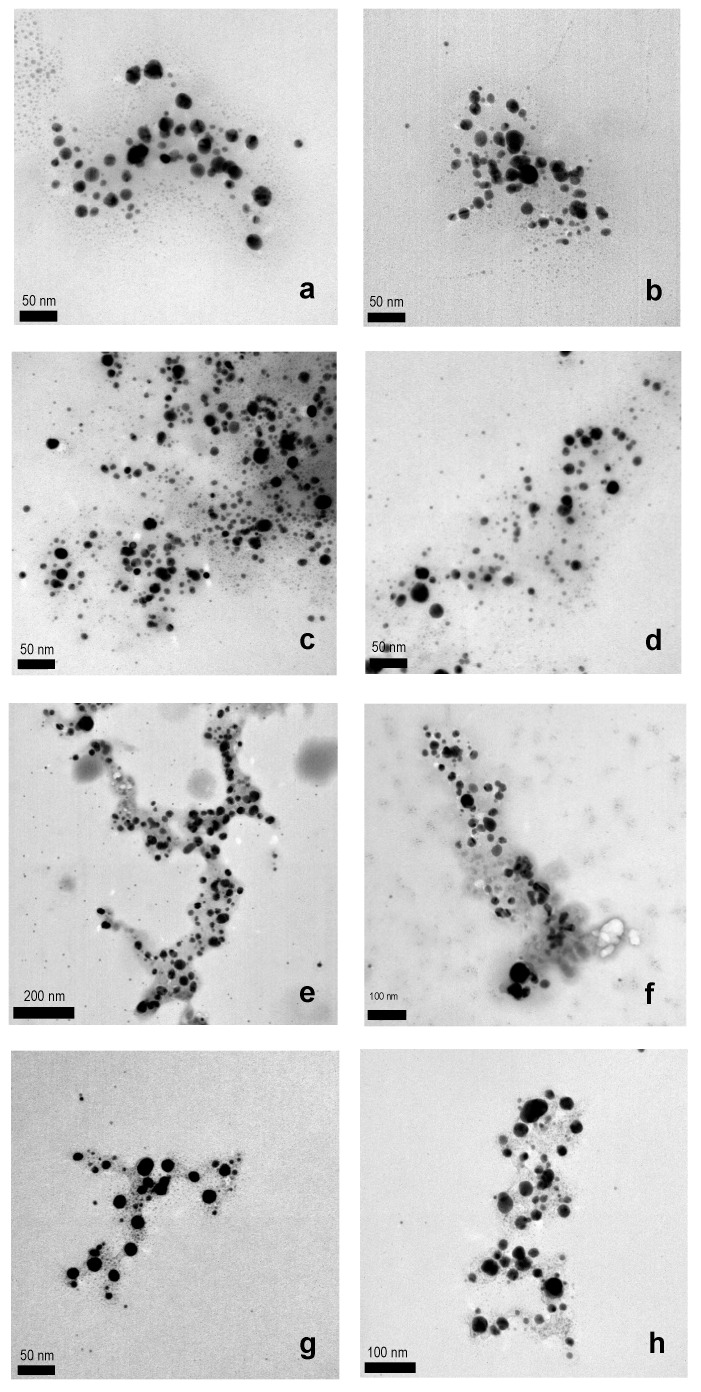
Representative TEM images of AgNPs synthesized by using WEOP of: (**a**,**b**) *Bellis perennis* petals, (**c**,**d**) *Allium sativum* peel, (**e**,**f**) *Allium porrum* green zone, and (**g**,**h**) *Allium porrum* white zone.

**Figure 3 nanomaterials-13-01010-f003:**
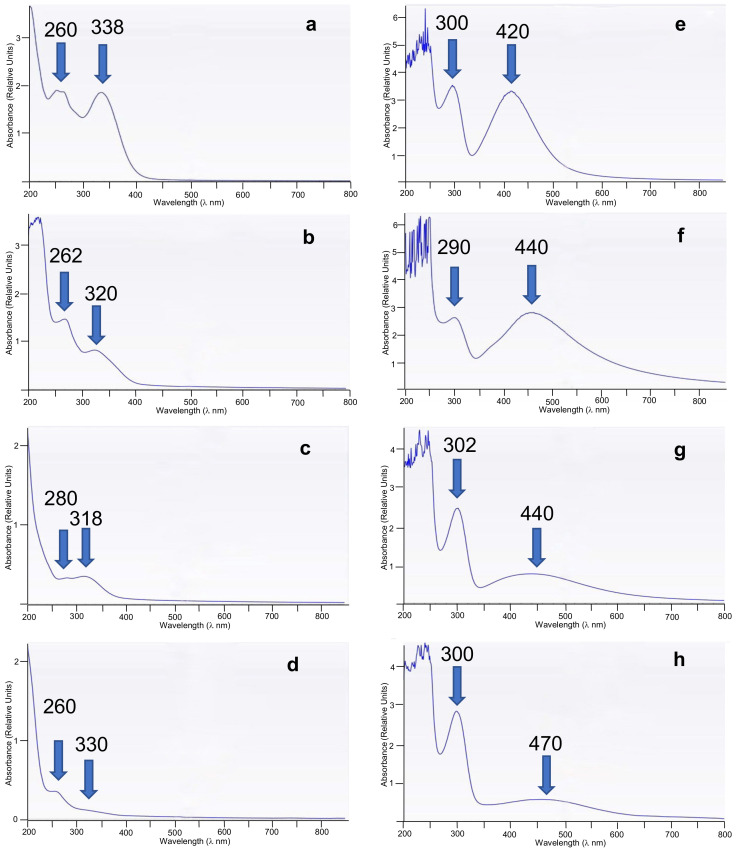
UV-Vis spectra (λ in nm) of WEOP of (**a**) *Bellis perennis* petals (daisy), (**b**) *Allium sativum* peel (garlic), (**c**) *Allium porrum* (leek) green zone, (**d**) *Allium porrum* white zone; and AgNPs prepared from WEOP of (**e**) *Bellis perennis* petals, (**f**) *Allium sativum* peel, (**g**) *Allium porrum* green zone, (**h**) *Allium porrum* white zone.

**Figure 4 nanomaterials-13-01010-f004:**
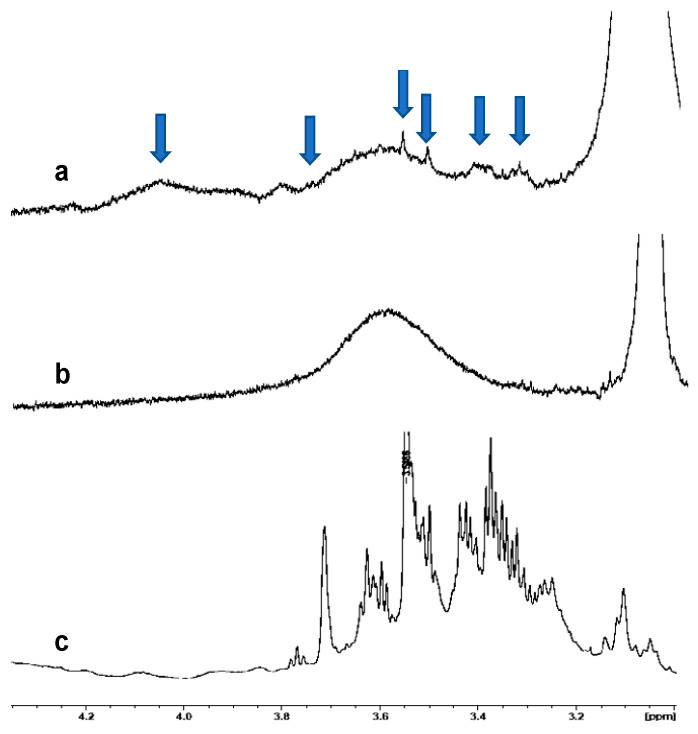
Comparison of the ^1^H-NMR spectra in DMSO-d_6_ of (**a**) PCE encapsulated in *p*NIPAM, (**b**) *p*NIPAM and (**c**) PCE.

**Figure 5 nanomaterials-13-01010-f005:**
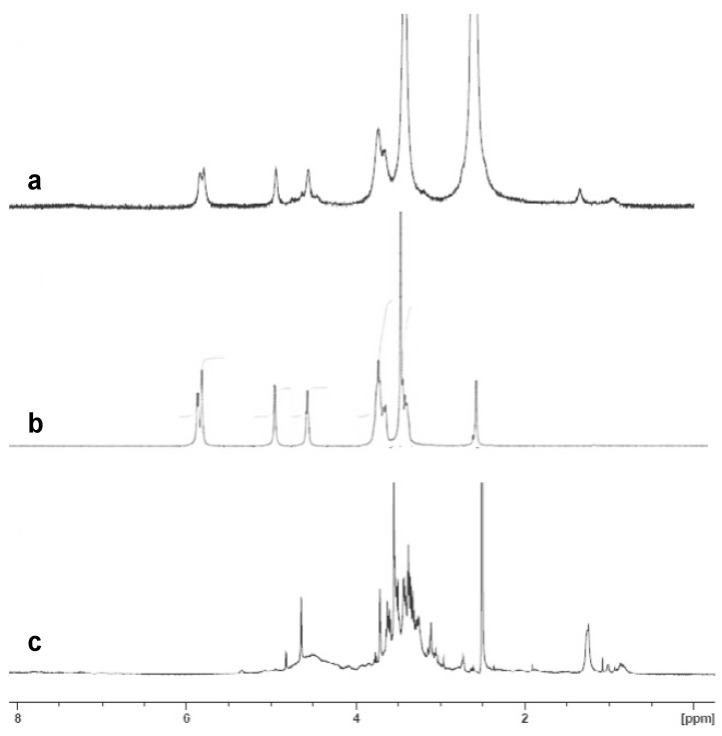
Comparison of the ^1^H-NMR spectra in DMSO-d_6_ of (**a**) PCE encapsulated in βCD, (**b**) βCD and (**c**) PCE.

**Figure 6 nanomaterials-13-01010-f006:**
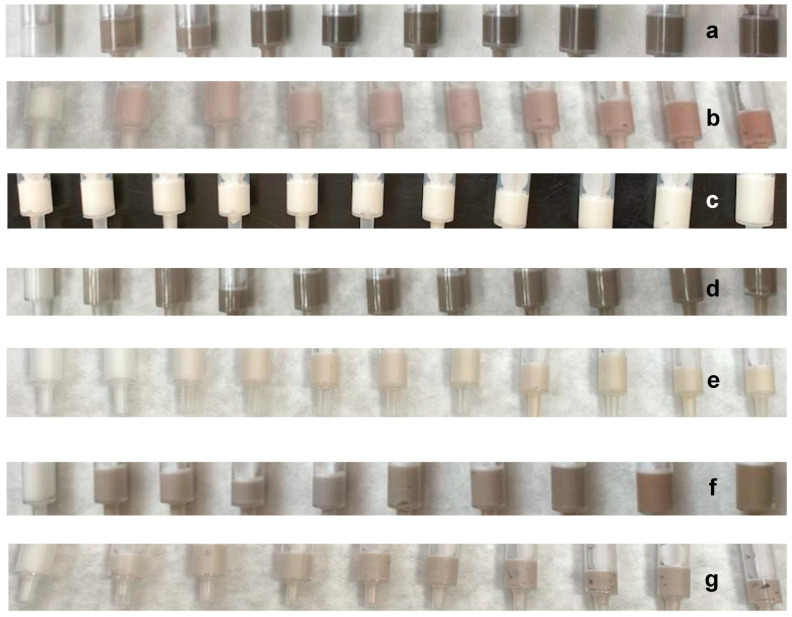
Body creams (1.0 g) prepared with (**a**) AgNPs (blank, 0.1, 0.2, 0.3, 0.4, 0.5, 0.6, 0.7, 0.8 and 0.9 mg), (**b**) PCE (blank, 0.1, 0.2, 0.3, 0.4, 0.5, 0.6, 0.7, 0.8 and 0.9 mg), (**c**) PCE after two days from addition, (**d**) AgNPs/PCE (blank, 0.1/1.0, 0.2/2.0, 0.3/3.0, 0.4/4.0, 0.5/5.0, 0.6/6.0, 0.7/7.0, 0.8/8.0, 0.9/9.0 and 1.0/10.0 mg), (**e**) PCE encapsulated in βCD (blank, 1.0, 2.0, 3.0, 4.0, 5.0, 6.0, 7.0, 8.0, 9.0 and 10.0 mg), (**f**) AgNPs and PCE encapsulated in βCD (blank, 0.1/1.0, 0.2/2.0, 0.3/3.0, 0.4/4.0, 0.5/5.0, 0.6/6.0, 0.7/7.0, 0.8/8.0 and 0.9/9.0 mg), and (**g**) PCE encapsulated in *p*NIPAM (blank, 1.0, 2.0, 3.0, 4.0, 5.0, 6.0, 7.0, 8.0, 9.0 and 10.0 mg). All images show increased concentration from left to right.

**Figure 7 nanomaterials-13-01010-f007:**
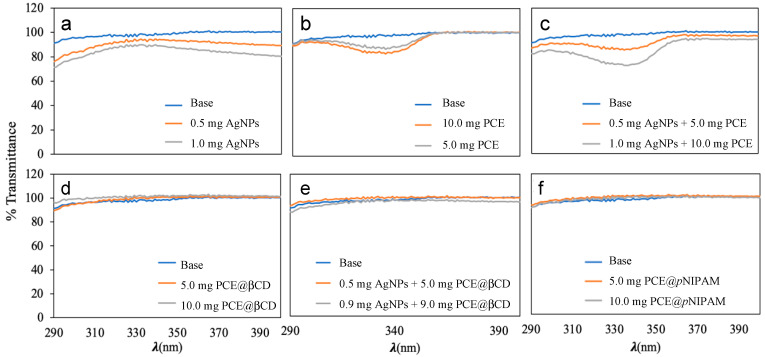
UV-Vis (λ 290–400 nm) transmittance plots for (**a**) body cream with AgNPs, (**b**) body cream with PCE, (**c**) body cream with AgNPs and PCE, (**d**) body cream with PCE encapsulated in βCD, (**e**) body cream with AgNPs and PCE encapsulated in βCD and (**f**) body cream with PCE encapsulated in *p*NIPAM. Values are expressed as transmittance percentage (%) and quantities in mg in 1 g of body cream.

**Table 1 nanomaterials-13-01010-t001:** UV-Vis data, quantities of AgNPs obtained and reaction yields.

Entry	WEOP	UV WEOP (λ_max_, log ε)	AgNPs (mg)	UV AgNPs (λ_max_, log ε)	Yield (%)	Shape & Size (nm Range & Average)
1	Daisy	264 (3.20), 338 (3.25)	783.271	300 (3.75), 420 (3.80)	84	Spherical (5–23, 16.1 ± 1.1)
2	Garlic	262 (3.10), 320 (2.70)	593.758	290 (3.70), 440 (3.79)	64	Spherical (5–25, 17.4 ± 1.7)
3	Leek green zone	280 (2.35), 318 (2.45)	348.437	302 (3.87), 440 (2.45)	38	Spherical (9–35, 18.3 ± 2.1)
4	Leek white zone	260 (2.25), 330 (2.01)	336.597	300 (3.97), 470 (2.40)	36	Spherical (5–25, 17.0 ± 2.0)

**Table 2 nanomaterials-13-01010-t002:** Total phenolic content (TPC) of the different plant species.

Entry	Sample	C (mg/L Gallic Acid)	V (mL WEOP)	W (mg Plant)	TPC (mg Gallic Acid/g Dry Extract)
1	Daisy	36.30 ± 0.50	89	0.425	7.60 ± 0.10
2	Garlic	7.76 ± 0.07	91	0.514	1.37 ± 0.10
3	Leek green zone	4.20 ± 0.40	93	1.018	0.38 ± 0.04
4	Leek white zone	3.53 ± 0.08	92	1.028	0.316 ± 0.007

**Table 3 nanomaterials-13-01010-t003:** Content of MAAs and phenolic compounds, and antioxidant capacity in the PCE.

MAAs (mg/g DE)	Phenols (mg/g DE)	Antioxidant Capacity (μmol TE/g DE)
30.3 ± 1.8	53.5 ± 3.5	32.5 ± 2.3

**Table 4 nanomaterials-13-01010-t004:** Sun protection factor (SPF), UV-A protection factor (UVAPF) and different biological effective protection factors (BEPFs) against other biological effects of the UV-A radiation (elastosis and photoaging). Values are expressed as mean ± standard deviation.

Entry		SPF	UVAPF	Elastosis	Photoaging
1	Base cream	1.03 ± 0.00	1.01 ± 0.00	1.01 ± 0.00	1.01 ± 0.00
2	0.5 mg AgNPs	1.13 ± 0.02	1.08 ± 0.01	1.09 ± 0.01	1.08 ± 0.01
3	1.0 mg AgNPs	1.20 ± 0.01	1.16 ± 0.01	1.17 ± 0.01	1.15 ± 0.01
4	5.0 mg PCE	1.07 ± 0.00	1.06 ± 0.00	1.05 ± 0.00	1.06 ± 0.00
5	10.0 mg PCE	1.10 ± 0.03	1.08 ± 0.02	1.07 ± 0.02	1.09 ± 0.03
6	0.5 mg AgNPs + 5.0 mg PCE	1.11 ± 0.01	1.09 ± 0.00	1.08 ± 0.00	1.09 ± 0.01
7	1.0 mg AgNPs + 10.0 mg PCE	1.21 ± 0.01	1.19 ± 0.01	1.17 ± 0.01	1.20 ± 0.01

## Data Availability

Archived data sets can be found at the Department of Organic Chemistry, University of Málaga.

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
