# Peer review of "Green Synthesis of Silver Nanoparticles and Its Combination with Pyropia columbina (Rhodophyta) Extracts for a Cosmeceutical Application"

_nanomaterials, 2023, doi:10.3390/nano13061010_

Round 1

Reviewer 1 Report

The topic studied by the authors is interesting and within the scope of the journal. The paper, however, cannot be published in the present form, and its objective(s) should be changed/redefined.

My main criticism is on the performances of these systems as photoprotectors. Spectra of Figure 7, in fact, show that the transmittance is very high, always above 70 %, for all formulations; indeed the SPF value is very low - just above 1. I would not define these as sunscreens. 

Based on these results, the target of the study should be changed completely - these may be good materials but surely they cannot be applied for photoprotection as stated in the title. Different functional properties should be explored for different applications.  

Other points: in the introduction, the authors talk about "physical, chemical and biological sunscreens" and this classification is wrong. What they define as "biological" are in reality chemical sunscreen of natural origin; talking about biological does not make sense. This is a major mistake for scientists who are supposed to be experts in the field.

Coming to the results: the legend for the dimension bars in TEM images should be larger, at the moment it is not possible to see what is written and which are the dimensions. 

For the UV spectra: it would be better to put in the same graph the spectra of each extract with that of the corresponding AgNPs, to make the comparison easier. 

Equation 2 should be in the Materials and methods section.  Also, regarding the cosmetic formulations, the authors just talk about a cream, without giving details, which should be included.

Author Response

We thank the referees for the comments. We have revised the manuscript according to these comments. Main changes are highlighted in yellow in the revised manuscript.  

Reviewer 1

The topic studied by the authors is interesting and within the scope of the journal. The paper, however, cannot be published in the present form, and its objective(s) should be changed/redefined.

My main criticism is on the performances of these systems as photoprotectors. Spectra of Figure 7, in fact, show that the transmittance is very high, always above 70 %, for all formulations; indeed the SPF value is very low - just above 1. I would not define these as sunscreens. 

Authors: the referee is right; the solar protection factor (SPF) and other protection factors (Elastosis and Photoaging) are low, with almost no photoprotection. Only samples with 1.0 mg AgNPs + 10.0 mg PCE we find some improvement; however, the measured values are close to 1.20, still very low. This last result pointed out a certain pickering effects of AgNPs as it is discussed. In any case, the low photoprotection factors measured are in concordance with the low incorporation of the active compounds into the formulated creams, with a maximum of 1 %.

Based on these results, the target of the study should be changed completely - these may be good materials but surely they cannot be applied for photoprotection as stated in the title. Different functional properties should be explored for different applications. 

Authors: we have deleted references on UV screen and photoprotection, and we have focused the  discussion on other properties of the extracts, as the antioxidant capacity, and we refer now in the new title as cosmeceutical applications, since antioxidants can reduce photoaging and other negative biological effects. Initially, after the successful design of the green synthesis of silver nanoparticles, we focused the research to improve the photoprotection capacity of the extracts by combination with the nanoparticles. This UV screen capacity of extracts of the red alga Pyropia columbina (previous Porphyra columbina) or purified mycosporine like amino acids from Porphyra, was previously demonstrated (De la Coba et al., 2019, Marine Drugs; Schneider et al., 2020, Algal Res.). However, since the concentration of extract/AgNPs used in this study was very low, also low photoprotection factors were measured: in the next studies more concentrated extracts and/or purified MAAs will be used.

Other points: in the introduction, the authors talk about "physical, chemical and biological sunscreens" and this classification is wrong. What they define as "biological" are in reality chemical sunscreen of natural origin; talking about biological does not make sense. This is a major mistake for scientists who are supposed to be experts in the field.

Authors: we named biological photoprotectors to the natural substances having photoprotection and antioxidant capacity. Different authors also used the term “biological filters” for UV screens and antioxidant substances. Certainly, in an extract with biological photoprotectors, it can be found both organic UV absorbers having specific chromophores, and inorganic compounds (that can be consider physical UV screen). Osterwalder et al. (Photodermatology, photoimmunology & photomedicine, 62 2014) reported as UV screen substances, inorganic UV filters and organic particulate UV filters produced by artificial chemical synthesis. We have prepared crude extracts from a red alga and in these extracts, there are natural molecules with both UV screen and antioxidant capacities (Mycosporyne like aminoacids) and also inorganic compounds that help in photoprotection. We have rewritten this part considering inorganic, organic UV filters (produced artificially by chemical synthesis) and natural organic UV filters with antioxidant capacity.

Coming to the results: the legend for the dimension bars in TEM images should be larger, at the moment it is not possible to see what is written and which are the dimensions. 

Authors: we have included larger dimension bars in TEM images.

For the UV spectra: it would be better to put in the same graph the spectra of each extract with that of the corresponding AgNPs, to make the comparison easier.

Authors: we decided to organize the UV spectra as in figure 3 in order to make the comparison among the UV spectrum of each extract (a-d) and that of AgNPs (e-h) easier. We think that putting both in the same graph will difficult the comparison, avoiding space for wavelength data.

Equation 2 should be in the Materials and methods section.

Authors: we have included equation 2 (now eq. 1) in Materials and Method section.

Also, regarding the cosmetic formulations, the authors just talk about a cream, without giving details, which should be included.

Authors: we have included the main composition for the body cream base used.

Reviewer 2 Report

When I read the manuscript I was a bit confused on the actual aim of the authors research, since they decided to present their findings in many different systems, without clearly stating why it was necessary to put them all together (it appears from the results that in fact it wasn't necessary at all). So, the reader doesn't really know whether he is going to have an insight on the green synthesis of silver NPs, or the use of algae extract as photoprotection additives, or the effect of encapsulation methods on such species' properties.

I wish the authors could give a clearer cut to their work, based on their most interesting results.

Anyway, even if the results are not spectacular, I believe they can be published after some revisions:

1) Title and abstract: I am not sure the title properly reflects the content of the paper, since the green synthesis has a relatively small part in it, while the abstract has the same flaws of the entire work: too many different systems are suggested but a clear idea of the innovative results is given;

2) Introduction: the reason for the specific choice of vegetal sources for the synthesis of NPs and of encapsulation systems for MAAs should be stated. What was the rationale in their selection amongst other possibilities?

3) Some further characterization of the NPs should be given. Have the authors measured the zeta potential? How was size measured? Size measurement error should be added in Table 1 and histograms provide in the supplementary.

4) In line 382 the authors state that NMR data in accordance with almost total incorporation of the PCE. How is that? No quantitative data are given.

5) I'd expect the so called PCE to be a mixture of different species, among which MAAs. Why are the authors so sure that the incorporation into pNIPAM and, more specifically, betaCD interests all those species? The provided NMRs are not enough.

6) The release of the MAAs species proved to take a relatively long time. No data on betaCD release have been provided. As shown encapsulation is detrimental to the devised application. Was the transmittance tested at different times?

7) The English language needs to be revised.

Author Response

We thank the referees for the comments. We have revised the manuscript according to these comments. Main changes are highlighted in yellow in the revised manuscript.

Reviewer 2

When I read the manuscript I was a bit confused on the actual aim of the authors research, since they decided to present their findings in many different systems, without clearly stating why it was necessary to put them all together (it appears from the results that in fact it wasn't necessary at all). So, the reader doesn't really know whether he is going to have an insight on the green synthesis of silver NPs, or the use of algae extract as photoprotection additives, or the effect of encapsulation methods on such species' properties.

I wish the authors could give a clearer cut to their work, based on their most interesting results.

Authors: the objective has been clarified. The initial idea was to design and prepare AgNPs by following green synthetic processes and to incorporate them into cosmetic formulations in combination with algal extracts with UV screen and antioxidant properties. The incorporation of algal extracts to the creams resulted in UV screen effect very low. Only in the case of formulations with 1.0 mg AgNPs + 10.0 mg PCE the values slightly improved, being close to 1.20. This last result pointed out a certain pickering effects of AgNPs as it is discussed. However, the combination of particles and extracts reports high antioxidant capacity for cosmeceutical applications.

Anyway, even if the results are not spectacular, I believe they can be published after some revisions:

1)Title and abstract: I am not sure the title properly reflects the content of the paper, since the green synthesis has a relatively small part in it, while the abstract has the same flaws of the entire work: too many different systems are suggested but a clear idea of the innovative results is given;

Authors: title has been changed to “Green synthesis of silver nanoparticles and its combination with Pyropia columbina(Rhodophyta) extracts for a cosmeceutical application”

2) Introduction: the reason for the specific choice of vegetal sources for the synthesis of NPs and of encapsulation systems for MAAs should be stated. What was the rationale in their selection amongst other possibilities?

Authors: we have chosen daisy petals (Bellis perennis), leek (Allium porrum) and garlic skin (Allium sativum) as plant materials, because to date, to the best of our knowledge, they have not been studied as reducing agents for metal nanoparticles preparation. We have included in the revised version of the manuscript a paragraph regarding this point. On the other hand, our group for many years has been working in encapsulation of natural active compounds into polymeric nanosystems a cyclodextrins. We observed that inclusion in these system produces an additional stabilization for the actives, also improving solubility in hydrophilic solvents. Because of this, we decided to test the behavior of MAAs under pNIPAM and betaCD systems.

3) Some further characterization of the NPs should be given. Have the authors measured the zeta potential? How was size measured? Size measurement error should be added in Table 1 and histograms provide in the supplementary.

Authors: NPs size was measured with the Java-based image processing program Image-J (software 3.8.9.21 version). Size measurement error has been added in Table 1, and statistical charts have been included in SM.

4) In line 382 the authors state that NMR data in accordance with almost total incorporation of the PCE. How is that? No quantitative data are given.

Authors: to clarify this point we have included one paragraph in 2.7 section: “To confirm the entrapment efficacy, the aqueous layer was decanted and then exhaustively extracted with dichloromethane (3 × 5 mL). The dichloromethane extracts were dried over MgSO4 and concentrated to dryness. The residue was dissolved in CDCl3 and analyzed by 1H-NMR.” Also in the 3.5 section: “Moreover, NMR technique shows values near to total incorporation of PCE into the NPs, since no MAAs signals are observed in the 1H-NMR spectra of the extracts of the aqueous layer obtained after centrifugation (see Experimental), and consequently the complete entrapment of PC actives by the polymeric matrix.” As mentioned in the manuscript, the quantitative incorporation of PCE into the NPs can be explained by the low PCE/polymer relationship during preparation, which is low enough to guarantee the complete incorporation of the PCE into the polymer matrix. We already used this methodology to ensure complete entrapment of active compounds by the polymeric nanoparticles (paclitaxel in Nano Res2017, 856).

5) I'd expect the so called PCE to be a mixture of different species, among which MAAs. Why are the authors so sure that the incorporation into pNIPAM and, more specifically, betaCD interests all those species? The provided NMRs are not enough.

6) The release of the MAAs species proved to take a relatively long time. No data on betaCD release have been provided. As shown encapsulation is detrimental to the devised application. Was the transmittance tested at different times?

Authors: as mentioned, incorporation of actives into pNIPAM nanoparticles was confirmed by NMR and FTIR. From NMR measurements of the aqueous layer obtained after centrifugation a complete entrapment can be induced. In case of betaCD we only check the formation of the inclusion complex (NMR and Raman), but no data for release or inclusion percentage was recorded. Due to the detrimental effect of betaCD-PCE complex on the photoprotective properties with decided not to go forward. However, it is common to find a similar incorporation pattern into de betaCD for molecules with similar chemical structures, so we find reasonable to assume a similar interest of all MAAs for the betaCD skeleton. Additionally, it is common the encapsulation of plant extracts in betaCD (for example: J. Polymer. Enviroment 2021, 2628; Trends in Food Sci. Technol.2021, 177; or Biosci. Biotechnol. Biochem. 2017, 718). In these cases, several actives with similar chemical structures are included into the betaCD structure.

7) The English language needs to be revised.

Authors: English has been improved.

Reviewer 3 Report

The manuscript reports the green synthesis of silver nanoparticles by using different biomass as reducing agents. The topic is of broad interesting. However, major revision is required before it could be accepted.

1.     Green synthesis of AgNPs is an interesting topic. Many typical researches have been done on this field. Some typical references are suggested to be cited, e.g. Journal of Bioresources and Bioproducts 2021, 6 (1), 75-81; Journal of Bioresources and Bioproducts 2021, 6 (1), 1-10.

2.     Please pay attention to the typos like “2.2. . Preparation of aqueous extract…”, “2.3. . Green synthesis…”, etc.

3.     It would be better to add a statistical chart to show the particle size distribution of AgNPs synthesized from different biomass reducant.

4.     Is it safe to add AgNPs into cosmetic formulations as body cream? Will AgNPs pass through skin? Did other literatures report similar researches?

5.     How about the photoprotective properties of as synthesized body cream compared to other counterparts?

6.     As a body cream, the antibacterial property, water retention capacity, etc. are also important. How about the performance?

Author Response

We thank the referees for the comments. We have revised the manuscript according to these comments. Main changes are highlighted in yellow in the revised manuscript.

Reviewer 3

The manuscript reports the green synthesis of silver nanoparticles by using different biomass as reducing agents. The topic is of broad interesting. However, major revision is required before it could be accepted.

  1. Green synthesis of AgNPs is an interesting topic. Many typical researches have been done on this field. Some typical references are suggested to be cited, e.g. Journal of Bioresources and Bioproducts 2021, 6 (1), 75-81; Journal of Bioresources and Bioproducts 2021, 6 (1), 1-10.

Authors: we thank the reviewer for this advice. We have included both references.

  1. Please pay attention to the typos like “2.2. . Preparation of aqueous extract…”, “2.3. . Green synthesis…”, etc.

Authors: these mistakes have been corrected.

  1. It would be better to add a statistical chart to show the particle size distribution of AgNPs synthesized from different biomass reducant.

Authors: statistical charts have been included in SM (SM-1).

  1. Is it safe to add AgNPs into cosmetic formulations as body cream? Will AgNPs pass through skin? Did other literatures report similar researches?

Authors: the use of silver nanoparticles in cosmetics formulations has already been reported (see for example, Molecules2023, 645, just to mention a recent article), and they are frequently used in pharmaceuticals preparations due to their antimicrobial and anti-inflammatory activity (Nanobiomaterials in Galenic Formulations and Cosmetics: Applications of Nanobiomaterials 2016, 391). Moreover, safety studies have also been reported, for example in Nanocosmeceuticals:Innovation, Application, and Safety 2022, 1–5721, among others.

  1. How about the photoprotective properties of as synthesized body cream compared to other counterparts?

Authors: the photoprotection capacity (both UV screen and antioxidant capacity) of algal extracts and body creams has been previously reported (De la Coba et al., Marine Drugs 2019; Schneider et al., Algal Res. 2020, Vega et al., Marine Drugs 2020). The values of SPF were higher than that in this study and in addition, as in this study, they present high antioxidant capacity.

  1. As a body cream, the antibacterial property, water retention capacity, etc. are also important. How about the performance?

Authors: other cosmetic properties are being evaluated in cosmetic creams with different algal extracts, including toxicity and allergic characteristics, among others, and will be included in future manuscripts. We have also demonstrated antiphotoaging capacity (inhibition of collagenase of mycosporine like amino acids of the extracts, Rodrigues Moreira et al., Algal Res. 2022). In addition, photo- and thermos-stability has been conducted with higher concentrated extracts if compared to that used in this study.

Round 2

Reviewer 1 Report

The authors addressed most of the points raised by the reviewers.

My only criticism is that they insist on talking about bilogical sunscreens, which is a mistake. In the reference they cite (Ostenwalder, Photoderm. Photoimm. Photomed.), there is never the use of this term. 

Therefore, in my opinion, for the paper to be published, they have to remove this wrong definition.

After that, the publication can go ahead.

Author Response

We have simplified the paragraph to remove the ambiguity.

Reviewer 3 Report

The manuscript has been revised according to the comments and suggest to be accepted.

Author Response

We thank the reviewer for his/her revision of the manuscript.